Differential mechanisms underlying responses of soil bacterial and fungal communities to nitrogen and phosphorus inputs in a subtropical forest

http://orcid.org/0000-0001-9146-8044 Li Yong 1
Tian Dashuan 2
http://orcid.org/0000-0002-3425-7387 Wang Jinsong 2
Niu Shuli 2
Tian Jing 2
Ha Denglong 3
Qu Yuxi 3
Jing Guangwei 3
Kang Xiaoming 1 xmkang@ucas.ac.cn
Song Bing 4 songbing@ldu.edu.cn
1 Beijing Key Laboratory of Wetland Services and Restoration, Institute of Wetland Research, Chinese Academy of Forestry , Beijing , China
2 Key Laboratory of Ecosystem Network Observation and Modeling, Institute of Geographic Sciences and Natural Resources Research , Beijing , China
3 Jigongshan Natural Reserve , Xinyang , China
4 School of Resources and Environmental Engineering, Ludong University , Yantai , China
Le Roux Xavier
Electronic publication date: 2019 Sep 6
Publication date: 2019
Volume: 7
Electronic Location ID: e7631
Received 2019 Apr 12; Accepted 2019 Aug 6
Copyright: © 2019 Li et al.
Copyright year: 2019
Copyright holder: Li et al.
License: This is an open access article distributed under the terms of the Creative Commons Attribution License, which permits unrestricted use, distribution, reproduction and adaptation in any medium and for any purpose provided that it is properly attributed. For attribution, the original author(s), title, publication source (PeerJ) and either DOI or URL of the article must be cited.
License URL: https://creativecommons.org/licenses/by/4.0/

Keywords: Nitrogen deposition, Phosphorus addition, Microbial diversity, Community composition, Amplicon sequencing, Subtropical forest, Soil bacteria and fungi

Funding: National Natural Science Foundation of China 41403073 and 31600356 Thousand Young Talents program This study was financially supported by National Natural Science Foundation of China (Grant Nos. 41403073 and 31600356) and the Thousand Young Talents program to Shuli Niu. The funders had no role in study design, data collection and analysis, decision to publish, or preparation of the manuscript.

==============================
Atmospheric nitrogen (N) deposition and phosphorus (P) addition both can change soil bacterial and fungal community structure with a consequent impact on ecosystem functions. However, which factor plays an important role in regulating responses of bacterial and fungal community to N and P enrichments remains unclear. We conducted a manipulative experiment to simulate N and P inputs (10 g N · m−2 · yr−1 NH4NO3 or 10 g P · m−2 · yr−1 NaH2PO4) and compared their effects on soil bacterial and fungal species richness and community composition. The results showed that the addition of N significantly increased NH4+ and Al3+ by 99.6% and 57.4%, respectively, and consequently led to a decline in soil pH from 4.18 to 3.75 after a 5-year treatment. P addition increased Al3+ and available P by 27.0% and 10-fold, respectively, but had no effect on soil pH. N addition significantly decreased bacterial species richness and Shannon index and resulted in a substantial shift of bacterial community composition, whereas P addition did not. Neither N nor P addition changed fungal species richness, Shannon index, and fungal community composition. A structural equation model showed that the shift in bacterial community composition was related to an increase in soil acid cations. The principal component scores of soil nutrients showed a significantly positive relationship with fungal community composition. Our results suggest that N and P additions affect soil bacterial and fungal communities in different ways in subtropical forest. These findings highlight how the diversity of microbial communities of subtropical forest soil will depend on future scenarios of anthropogenic N deposition and P enrichment, with a particular sensitivity of bacterial community to N addition.

Introduction

Nitrogen (N) and phosphorus (P) inputs to the ecosystem have greatly increased due to anthropogenic activities, mainly originating from fossil fuel combustion, agricultural fertilization, and dust or ash production (Galloway et al., 2008; Stevens, 2019; Wang et al., 2014). Excessive N and P inputs could result in many adverse impacts, including soil acidification (Guo et al., 2010; Mao et al., 2017; Tian & Niu, 2015) and nutrient imbalance (Peñuelas et al., 2013), although N and P have been considered limiting factors for plant growth (Harpole et al., 2011). Moreover, elevated N availability may aggravate P limitation of ecosystem productivity (Li, Niu & Yu, 2016). A large body of research focusing on aboveground plant community responses have demonstrated multiple effects with nutrient additions, including biodiversity loss (Hooper et al., 2012), species composition, and associated ecosystem services, such as terrestrial carbon dynamics (Isbell et al., 2013; LeBauer & Treseder, 2008). However, the responses of belowground microorganisms to nutrient additions, including general taxonomic traits and related function shifts, remain unclear (Leff et al., 2015; Ma et al., 2016, 2019; Wei et al., 2018). In particular, integrated field experimental investigations of both bacterial and fungal responses to N and P additions are needed to improve our understanding of how the soil microbial community structure shifts in response to nutrient addition and whether the bacterial and fungal responses are consistent.

Soil microbial communities are sensitive to N enrichment (Dai et al., 2018; Ramirez, Craine & Fierer, 2012; Treseder, 2008). On average and based on a global meta-analysis, across all of the studies in agro-ecosystems, the abundances of Proteobacteria and Actinobacteria significantly increased by 2.2% and 1.1%, respectively, and the abundance of Acidobactria decreased by 2.3% under N addition, which suggest differential sensitivity of bacterial phyla to N inputs (Dai et al., 2018). N addition could lead to elevated N availability (Chen et al., 2019), ammonium toxicity (Van Den Berg et al., 2005), acid cation toxicity (Chen et al., 2016; Tian & Niu, 2015), and loss of base mineral cations (Bowman et al., 2008). Meanwhile, several potential driving factors were proposed to account for shifts in sensitive microbial communities in some ecosystems (Nie et al., 2018). First, soil pH has been reported as a strong predictor of soil bacterial community composition at the continental scale (Lauber et al., 2009; Rousk et al., 2010). A significantly positive correlation between diversity of bacteria and soil pH might be attributable to narrow pH ranges for optimal growth of bacteria (Rousk et al., 2010). However, fungal community composition is less affected by soil pH because fungi generally exhibit wider pH ranges for optimal growth (Rousk et al., 2010). Second, N addition could directly affect soil bacterial community composition through modified ammonium N concentration (Nie et al., 2018; Zeng et al., 2016; Zhou et al., 2017). Soil bacterial community composition is closely related to soil NH4+-N content in tropical forest soil under N enrichment due to a narrow pH decrease in severely acidic soil (pH < 4.5) (Nie et al., 2018). Finally, a positive relationship between plant and fungal beta diversity has been reported under N enrichment, soil properties including soil inorganic N, pH, and associated extractable cations being correlated with compositional changes in plant and fungal communities (Chen et al., 2018b). Although those mechanisms have previously been investigated, comprehensive studies on different regulatory pathways of N addition on bacterial and fungal community structure in forest ecosystems are limited.

Phosphorus availability plays an important role in affecting microbial growth; however, our understanding of soil microbial community responses to elevated P inputs remains limited (Leff et al., 2015; Ma et al., 2016, 2019). Moreover, the effects of P input on bacterial and fungal community composition can differ (Cassman et al., 2016; Jorquera et al., 2014; Liu et al., 2012b, 2013). Previous studies have demonstrated that P enrichment induces shifts in microbial community composition by increasing P availability and altering soil and plant chemistry (DeForest & Scott, 2010; Lagos et al., 2016; Leff et al., 2015; Liu et al., 2012a, 2013; Teng et al., 2018). Other studies have reported that only N enrichment influences bacterial abundance and community composition, whereas P input did not because soil pH decreases only with N addition (Jorquera et al., 2014; Wang et al., 2018) or because bacterial community was limited by other resources (Ma et al., 2019). The abovementioned pathways have largely been studied separately, and the relative contributions of these pathways to N- and/P-induced changes in bacterial and fungal communities, however, have not been investigated using field experiments.

Subtropical forests in southern China undergo extensive N deposition with signs of N saturation, such as soil acidification (Zhu et al., 2015). Previous studies have demonstrated that forest productivity and soil respiration are sensitive to N and P enrichment, but little is known about how P addition affects microbial communities (Li et al., 2018a, 2018b; Yu et al., 2017). To better understand the responses of bacterial and fungal communities to N and P addition and the underlying mechanisms, we conducted a N and P addition experiment in this region. We specifically addressed the following questions: (1) how do bacterial/fungal taxa and community structures respond to N and P enrichment? (2) How do soil biotic and abiotic factors regulate the responses of bacterial/fungal taxa and community structures to N and P enrichment? Finally, (3) what are the mechanisms underlying shifts in bacterial/fungal taxa and community structures to N and P enrichment?

Materials and Methods

Site description

This N and P addition experiment was setup at the Jigongshan Nature Reserve, China (31°51′58″N, 114°5′12″E). The site is experiencing northern subtropical to warm temperate climates because it is located within a climate-transitional region. The mean annual surface air temperature and mean annual rainfall at the reserve is 15.2 °C and 1,118 mm, respectively. The soil type is yellow-brown and soil thickness is about 0.3–0.6 m (Yan et al., 2014). The forest type is deciduous oak mixed forest. The predominant tree species in the canopy layer include Quercus acutissima and Q. variabilis and Liquidambar formosana, Lindera glauca, and Viburnum dilatatum dominates the understory arborous layer. The age of the stand is about 45–50 years and it is a secondary forest due to harvest in the late 1950s (Xu & Liang, 1965). The total N deposition in this region is about 30 kg N ha−1 yr−1 (Zhu et al., 2015). The soil properties at the beginning of the experiment is listed in Table S1.

Experimental design

The detailed information of N and P addition experiment could be found in previous studies (Li et al., 2018a, 2018b). Briefly, this N and P addition experiment was setup in July 2013 using a complete randomized block design. Each block had four treatments that were randomly assigned to 10 × 10 m plots and four replicate blocks were established. Control (CK, without N and P addition), N addition (N, 10 g m−2 yr−1 NH4NO3), P addition (P, 10 g m−2 yr−1 NaH2PO4), and NP addition (NP, 10 g · m−2 · yr−1 NH4NO3 + 10 g · m−2 · yr−1 NaH2PO4) were included. Backpack sprayer were used to spray 50 L of water dissolved additions for each plot onto the forest floor monthly from May to October each year. The control plot received 50 L of water (equivalent to 0.5 mm precipitation) each time.

Soil properties

Six soil cores (0–10 cm depth) were randomly collected from each plot and mixed to obtain one composite sample in October 2017. The samples were passed through a two mm sieve and three parts were divided. One part of the soil was used for the analysis of ammonium (NH4+), nitrate (NO3−), microbial biomass carbon and nitrogen (MBC and MBN), and dissolved organic carbon (DOC). The second part of fresh soil was collected in a 50-mL centrifuge tube, which was stored at −80 °C for soil DNA extraction. The remaining soil was air-dried for the determination of soil pH, total soil organic carbon (SOC), total nitrogen (TN), available phosphorus (AP), and extractable cations including Al3+, Ca2+, Mg2+, and Na+. The plant fine roots collected from six soil cores using the two mm-sieving were washed, dried, and then weighted. Soil ammonia and nitrate concentrations were determined by colorimetric analysis on a FIAstar 5000 Analyzer (FIAstar 5000 Analyzer; Foss Tecator, Hillerød, Denmark). Soil DOC was analyzed using a TOC analyzer (multi N/C 3100; Analytik Jena, Jena, Germany). Soil MBC and MBN were determined by a chloroform fumigation extraction method (Brookes et al., 1985). Soil pH was analyzed in a soil water solution (1:2.5 w/v). SOC and TN were analyzed by a C/N analyzer (vario EL III, CHNOS Elemental Analyzer; Elementar, Langenselbold, Germany). Available P was determined following by molybdenum blue colorimetry (Jin et al., 2019). A modified extraction procedure was used to measure extractable cations including Al3+, Ca2+, Mg2+, and Na+ (Rauret et al., 2000).

Soil DNA extraction, PCR amplification, and sequencing

Soil DNA was extracted from each sample using a Fast DNA Stool Mini Kit (Tiangen Biotech Beijing Co., Ltd., Beijing, China) according to the manufacturer’s instructions. The quality of the purified DNA was assessed based on the 260/280 and 260/230 nm absorbance ratios obtained using a NanoDrop ND-1000 spectrophotometer (NanoDrop Technologies Inc., Wilmington, DE, USA).

Microbial community diversity and composition were assessed by amplification of the 16S rRNA gene for the bacteria and the internal transcribed spacer (ITS) region of the fungi, as described in Prober et al. (2015). Marker genes in the isolated DNA were PCR-amplified and barcoded in triplicate reactions for both the 16S rRNA gene (using the 515f/907r primer pair) and the ITS1 region (using the ITS1F/ITS2R primer pair). Sequencing was conducted on an Illumina Miseq platform at Majorbio Biopharm Technology Co., Ltd., Shanghai, China.

Sequencing data accession numbers

Raw sequences were trimmed of reads containing ambiguous bases and long homopolymers and merged using QIIME v 1.7.0 (Caporaso et al., 2012). All filtered sequences from 16S rRNA and ITS gene amplicons were clusted into Operational Taxonomic Units (OTU) at 97% similarity cutoff using UPARSE (version 7.1, http://drive5.com/uparse), followed by chimera filtering usingn the ribosomal database project (RDP) (Tian et al., 2018) and UCHIME (Edgar et al., 2011). The bacterial and fungal OTU sequences were classified using the RDP classifier against Greengenes and UNITE reference database, repectively. All of the raw sequencing data (.fastq files) were submitted to the sequence read archive at the National Center for Biotechnology Information under accession number PRJNA531787.

Statistical analysis

The bacterial and fungal richness was determined by the number of OTUs, and alpha diversity was estimated using the Shannon index. Linear regressions were applied to assess the relationships between bacterial and fungal Shannon index and richness and environmental factors. Stepwise multiple regressions were used to identify the most influential environmental variables on bacterial and fungal Shannon index and richness due to collinearity among environmental factors. Differences in microbial species richness, Shannon index, soil properties and fine root biomass among different treatments were determined by one-way ANOVA with Duncan test. To determine the effect size of N, P and combined NP treatment on the relative abundances of the dominant bacterial and fungal taxa, the response ratio was calculated as ln (Xt/Xc), where Xt is the mean value of experimental treatment and Xc is the mean value of the control treatment. One-sample t-test was used to estimate whether each response ratio was significantly different from zero (She et al., 2018).

Principal component analysis (PCA) and nonmetric multidimensional scaling (NMDS) were used to determine changes in the bacterial and fungal communities. Analyses, including Anosim, Adonis, and MRPP, were further performed to assess the significant differences in community among different treatments. The partial Mantel test was used to evaluate the linkages between the microbial community structure and the environmental variables. Pairwise taxonomic distance between microbial communities (Bray–Curtis) and Euclidean distance of environmental variables were chosen for partial Mantel. PCA, NMDS, and partial Mantel were conducted using the vegan package with R v3.5.0 (R Core Team, 2018).

Structural equation modeling (SEM) was performed to analyze the hypothetical pathways of N and P addition effects on bacterial and fungal diversity and community composition. Before the SEM analysis, soil N availability including NH4+ and NO3−, soil acid cations including H+ and Al3+, soil nutrients including SOC, TN, DOC, MBC and MBN, and microbial community composition (OTUs) were subjected to PCA (Chen et al., 2013). Soil N availability, soil acidity, soil nutrients, and microbial community composition were presented by the first principal components (PC1) in the following SEM analysis. Maximum likelihood estimation method was applied in the process of SEM. The goodness of the models was determined by χ2 tests, Akaike information criteria (AIC), and root square mean errors of approximation (RMSEA) (Li et al., 2018a).

Results

Fine root biomass and soil properties

Nitrogen, P, and combined NP additions significantly reduced fine root biomass (Table S2). Soil pH decreased with N addition, whereas P and NP additions did not have any effect compared to the control. Soil NH4+ was increased by 99.6% under N addition, whereas it did not change after P and NP additions (Table S2). P and NP additions resulted in a significant increase in soil AP, but N addition did not (Table S2). Soil Al3+ increased by 57.4%, 27.0%, and 59.1%, respectively, under N, P and NP additions. Soil Ca2+ decreased by 33.6% and 25.4%, respectively, under N and NP additions, but did not significantly change under P addition. N, P, and NP additions did not significantly change SOC, total nitrogen (TN), ratio of SOC to TN (C/N ratio), DOC, MBC and MBN, NO3−, Mg2+, and Na+.

Relative abundance of dominant microbial taxa

A total of 16,573 to 26,745 (average: 21,019) and 46,414–70,354 (average: 64,655) valid sequences were consequently obtained per sample for bacterial and fungi, respectively. These sequences were grouped into 1,666 and 2,487 OTUs at the 97% similarity level for bacterial and fungi, respectively. All of the samples were compared at an equivalent sequencing depth of 16,573 and 46,414 per sample for bacteria and fungi, respectively.

The predominant bacterial phyla across all of the samples were Proteobacteria, Acidobacteria, and Actinobacteria (mean relative abundance > 5%), which accounted for more than 82% of the bacterial sequences on average (Fig. 1A). In addition, Planctomycetes, Chloroflexi, Firmicutes, Bacteroidetes, and Gemmatimonadetes were also present at relatively low abundance. The dominant fungal phyla across all of the samples were Ascomycota, Basidiomycota, Zygomycota, and Rozellomycota (Fig. 1B).

Figure 1 Relative abundance of the dominant bacterial (A) and fungal (B) groups at the phylum level under different treatments.

At the phylum level, Omnitrophica, Chlorobi, and Nitrospirae presented significant decrease and Saccharibacteria increased in relative abundance under N treatment (Fig. 2A). P addition significantly increased the relative abundances of Elusimicrobia, but N addition had no effect compared to the control plots (Fig. 2A). Only NP treatment increased the relative abundance of Firmicutes and decreased that of Chloroflexi (Fig. 2A). No significant differences were observed in the fungal relative abundance of Ascomycota, Rozellomycota, and Chytridiomycota at the phylum level among different treatments (Fig. 2B). Basidiomycota presented significant decrease in their relative abundance under N and NP additions and only NP addition decreased the relative abundance of Zygomycota (Fig. 2B). At the fungal genus level, Chaetomium and Purpureocillium presented significant decrease in their relative abundance under N and P treatments, and N and NP treatments decreased the relative abundance of Mycoarthris (Fig. 2B). The relative abundance of Mortierella significantly decreased only under NP addition (Fig. 2B).

Figure 2 Responses ratio analysis of changes in the relative abundance of dominant bacterial phyla (A) and fungal phyla/genera (B) in response to N, P and NP treatment compared to the control treatment at the 95% confidence interval.

Red points indicate significant changes compared with the control treatment.

Bacterial and fungal α diversity

Nitrogen addition significantly decreased bacterial α diversity, including phylotype richness (OTU numbers) and Shannon diversity index, but did not change fungal α diversity (Fig. 3). P addition had no effect on both bacterial and fungal α diversity. NP addition only decreased fungal phylotype richness (Fig. 3). N addition decreased the number of unique phylotypes compared to the control (Table S3).

Figure 3 Bacterial and fungal phylotype richness (A) and (B) and Shannon diversity index (C) and (D) under different treatments.

The error bars represent the SE of the mean (n = 4). Different letters above the bars indicate significant difference at P < 0.05.

The bacterial richness exhibited the largest correlation with soil pH among all of the environmental factors (Table 1, r = 0.80, P < 0.01). In addition to DOC (r = −0.72, P < 0.01), soil pH was the best predictor of bacterial diversity (Table 1, r = 0.71, P < 0.01). Bacterial richness and diversity also significantly correlated with fine root biomass, SOC, MBC, Al3+, and Ca2+ (Table 1). Soil Al3+ (r = −0.62, P < 0.05) presented the largest correlation with fungal richness, followed by soil pH (r = 0.57, P < 0.05). No significant correlations were noted between fungal diversity and any of the environmental variables (Table 1).

Table 1 Pearson correlations between bacterial and fungal richness and diversity and plant and soil characteristics.

r	Bacteria	Fungi	
Richness	Diversity	Richness	Diversity	
Fine root biomass	0.59*	0.56*	0.51*	0.32	
pH	0.80**	0.71**	0.57*	0.41	
SOC	−0.52*	−0.40	−0.29	−0.47	
TN	−0.47	−0.35	−0.16	−0.33	
C:N ratio	−0.40	−0.35	−0.42	−0.47	
DOC	−0.75**	−0.72**	−0.32	−0.20	
NH4+	−0.48	−0.34	−0.23	−0.17	
NO3−	−0.40	−0.31	−0.15	−0.02	
AP	0.07	0.24	−0.34	−0.42	
MBC	−0.64**	−0.49	−0.54*	−0.48	
MBN	−0.41	−0.27	−0.40	−0.42	
Al3+	−0.54*	−0.58*	−0.62*	−0.33	
Ca2+	0.58*	0.54	0.40	0.33	
Mg2+	0.24	0.09	0.22	0.13	
Na+	−0.06	−0.07	−0.33	−0.17	
Notes:

* 0.01 < P ≤ 0.05.

** 0.001 < P ≤ 0.01.

SOC, soil organic carbon; TN, total nitrogen content; DOC, dissolved organic carbon; AP, available phosphorus; MBC, microbial biomass carbon; MBN, microbial biomass nitrogen.

Stepwise regression showed that bacterial richness significantly correlated with pH (64.3%) and DOC (7.1%) (Table S4). Soil pH explained the largest part of the variation in bacterial diversity (50.1%), followed by TN (14.9%), DOC (13.4%), and MBC (9.7%). Soil pH explained 33.0% of the variation in fungal richness, whereas pH (17.0%), SOC (19.1%), and MBC (41.4%) collectively contributed to 65.2% of the variation in fungal diversity (Table S4).

Bacterial and fungal community structure and relationships with environmental factors

Significant differences in bacterial community structure between N and NP additions and the control were observed in NMDS and PCA plots (Fig. 4; Fig. S1). However, only NP addition exhibited significant differences in fungal community structure compared to the control (Fig. 4). The results of anosim, adonis, and MRPP analyses further confirmed the significant differences among treatments (Table S5). This finding was confirmed by a cluster analysis based on a Bray–Curtis distance matrix of soil samples under different treatments (Fig. S2).

Figure 4 Non-metric multidimensional scaling (NMDS) ordination of the soil bacterial (A) and fungal (B) community structure under different treatments.

The Bray–Curtis distance matrix based on the abundance of OTUs was used to determine the compositional variation.

A partial Mantel test was performed to reveal the major environmental variables shaping microbial community structure. pH (rM = 0.57, P = 0.001), Al3+ (rM = 0.40, P = 0.002) and fine root biomass (rM = 0.36, P = 0.005) were the most important factors that independently contributed to variations in soil bacterial community structure (Table 2). TN, C:N ratio, DOC, and MBC showed weak but significant correlations (rM = 0.21, rM = 0.22, rM = 0.27, rM = 0.24, respectively, P < 0.05) (Table 2). pH (rM = 0.57, P = 0.001) and MBC (rM = 0.40, P = 0.004) were the most important factors that independently contributed to variations in soil fungal community structure (Table 2). C:N ratio, DOC, Al3+, and Mg2+ showed weak but significant correlations (rM = 0.25, rM = 0.26, rM = 0.21, rM = 0.27, respectively, P < 0.05) (Table 2).

Table 2 Partial mantel test of soil bacterial and fungal community structure with environmental characteristics.

	Bacteria	Fungi	
	r	P	r	P	
Fine root biomass	0.36	0.005	0.10	0.18	
pH	0.57	0.001	0.40	0.001	
SOC	0.16	0.10	0.19	0.07	
TN	0.21	0.04	0.12	0.14	
C:N ratio	0.22	0.04	0.25	0.02	
DOC	0.27	0.02	0.26	0.01	
NH4+	0.08	0.21	−0.009	0.49	
NO3−	0.07	0.26	0.02	0.45	
AP	−0.14	0.95	−0.13	0.92	
MBC	0.24	0.03	0.36	0.004	
MBN	0.10	0.15	0.17	0.06	
Al3+	0.40	0.002	0.21	0.03	
Ca2+	0.12	0.18	−0.06	0.66	
Mg2+	0.06	0.28	0.27	0.02	
Na+	−0.15	0.91	−0.03	0.62	
Notes:

The correlation and significance were determined between bacterial and fungal community structure (Bray–Curtis distance) and environmental variables (Euclidean distance) based on 999 permutations. The bold numbers indicate significant correlations.

SOC, soil organic carbon; TN, total nitrogen content; DOC, dissolved organic carbon; AP, available phosphorus; MBC, microbial biomass carbon; MBN, microbial biomass nitrogen.

Pathways determining the responses of bacterial and fungal diversity and community composition

The SEM explained 12% of the variations in soil N availability, 83% of the variation in soil acid cations (H+ and Al3+), 5% of the variation in soil nutrients and 80% of the variation in soil available P (Fig. 5). The total variation in bacterial Shannon index was mainly explained (64%) by soil acid cations and soil available P (Fig. 5A), but the total variation in fungal Shannon index was mainly explained (39%) by soil available P (marginally significant, P = 0.08, Fig. 5B). The total variation in bacterial community composition was mainly explained (79%) by soil acid cations (Fig. 5A), but the total variation in fungal community composition was mainly explained (45%) by soil nutrients (Fig. 5B). Linear regression demonstrated that principal component (PC1) scores of fungal community composition was significantly correlated with PC1 scores of soil nutrients (R2 = 0.41, P = 0.008, Fig. S3).

Figure 5 SEM analysis of the effects of N and P enrichment on soil bacterial (A) and fungal (B) community structure.

Results of model fitting: (A) χ2 = 19.38, df = 13, P = 0.11, RMSEA = 0.18, AIC = 65.38; (B) χ2 = 18.39, df = 13, P = 0.14, RMSEA = 0.17, AIC = 64.39. Black and blue arrows represent significant positive and negative pathways, respectively, and dashed arrows indicate nonsignificant pathways. Numbers at the arrows are standardized path coefficients and arrow width is proportional to the strength of the relationship. R2 values on top of response variables indicate the proportion of variation explained by relationships with other variables. Prior to SEM analysis, soil N availability (NH4+ and NO3−), soil acid cations (H+ and Al3+), soil nutrients (SOC, TN, DOC, MBC and MBN), bacterial and fungal community composition (OTUs) were subject to PCA procedure to reduce the number of variables.

Standardized total effects from SEM demonstrated that N enrichment had stronger effects on bacterial diversity and community composition compared to P addition (Fig. 6). Specifically, acid cations showed the most powerful negative effect on bacterial diversity (Fig. 6A) and had a stronger positive effect on bacterial community composition than soil N availability (Fig. 6B). Both N and P enrichment had negative effects on fungal diversity, and soil nutrients exerted the most powerful positive effect on fungal diversity (Fig. 6C). N enrichment had stronger positive effects on fungal community composition, and soil nutrients exerted strongest negative effect on fungal community composition (Fig. 6D).

Figure 6 SEM-derived standardized total effects of N and P additions, and variables including N availability, acid cations, soil nutrients, and soil available P on bacterial diversity and community composition (A–B) and fungal diversity (C–D).

Discussion

N but not P addition significantly influences bacterial communities

Our results showed that N addition significantly decreased bacterial OTU richness and Shannon index and altered the bacterial community composition, with Saccharibacteria increasing, but Chloroflex decreasing in terms of their relative abundances (Figs. 2–4; Table S5). These results indicate that N addition tends to favor the copiotrophic phylum (Saccharibacteria) and counterselects the oligotrophic phylum (Chloroflex) in bacterial communities, which is in line with the copiotrophic hypothesis in previous reports (Fierer, Bradford & Jackson, 2007; Fierer et al., 2012; Leff et al., 2015). However, N addition did not change the relative abundance of dominant phylum of Proteobacteria, Acidobacteria, and Actinobacteria (Fig. 2). The results are in contrast with the findings of previous studies, which showed that added N stimulates and decreases the relative abundances of Proteobacteria (copiotrophic) and Acidobacteria (oligotrophic), respectively (Chen et al., 2019; Fierer et al., 2012; Leff et al., 2015). This discrepancy could be attributable to neutral effects of N enrichment on soil available C and N (including DOC and NO3−, Table S2) (Turlapati et al., 2013). It has been well documented that Proteobacteria contains N-fixing and N transforming genera (e.g., Bradyrhizobium, Burkholderia, Magnetospirillum, and Mesorhizobium) and thus are closely related to N cycling. However, in our study, added N only increased NH4+ and did not affect NO3−, indicating the N transforming-related bacterial phylum (Proteobacteria) may not be affected by the addition of N. Additionally, Acidobacteria has been reported to be negatively correlated with soil C availability (Fierer, Bradford & Jackson, 2007). The neutral effect of elevated N on DOC in our study may have contributed to the unaltered relative abundance of Acidobacteria, which is favored by low C availability (Fierer, Bradford & Jackson, 2007; Turlapati et al., 2013). Finally, the absence of a decrease in the abundance of Actinobacteria may be attributed to its relatively high tolerance to environmental stress such as low pH (Dai et al., 2018).

In contrast to most previous findings (Ling et al., 2017; Tan et al., 2013), neither the bacterial diversity nor the community composition was substantially influenced by P addition in our study (Figs. 3 and 4; Table S5), but this finding is consistent with a previous study reporting that most bacterial taxa were not limited by P in Tibetan meadow (Ma et al., 2019). Among all of the bacterial phyla identified, only Elusimicrobia were negatively correlated with soil available P (Table S6). There are several potential explanations for the insensitive response to added P. First, it is well established that nutrient addition-induced pH decline is an important mechanism in shifting the bacterial community composition (Chen et al., 2015; Ling et al., 2017; Zeng et al., 2016). However, P addition did not change the soil pH in our study (Table S2). Additionally, P input could increase the P availability, and thus affect the bacterial richness and community structure, especially in P-limited ecosystems (Li et al., 2015; Liu et al., 2013). Our results showed that increased soil P availability did not change microbial biomass, suggesting that microbial growth is not limited to P in this subtropical forest. Finally, Liu et al. (2013) pointed out that a decrease in labile SOC is associated with alterations in soil microbial community structure under P addition. In our study, however, neither SOC or DOC was not significantly influenced by P addition (Table S2). Overall, our results suggest that only increased P availability may not be able to change the bacterial community composition in subtropical forest.

Neither N nor P addition significantly influenced fungal community

Our results demonstrated that fungal OTU richness is positively correlated to fine root biomass and soil pH and negatively related to MBC and soil Al3+. However, fungal Shannon index was not significantly associated with soil parameters (Table 1). Fungal community structure was closely related to soil pH and MBC, followed by soil C:N ratio, DOC, Al3+, and Mg2+ (Table 2). In contrast to the bacterial community, both N and P additions did not significantly change fungal richness and diversity (Fig. 3), which suggests that the tolerances of fungi to environmental changes are generally stronger than those of bacteria (Fierer et al., 2012; Rousk et al., 2010). The responses of fungal diversity to N enrichment is not always consistent due to ecosystem type and nutrient dose (Chen et al., 2019; Mueller, Balasch & Kuske, 2014). Some studies have demonstrated that experimental N deposition results in increased fungal richness and diversity in a low fertility loblolly pine forest (Mueller, Balasch & Kuske, 2014; Weber, Vilgalys & Kuske, 2013), but other studies showed the opposite response partly due to variations in nutrient availability (Chen et al., 2018a, 2019). Previous studies have demonstrated that N enrichment increases the relative abundance of copiotrophic phylum Ascomycota and decreased that of oligotrophic phylum Basidiomycota according to the copiotroph-oligotroph concept (Chen et al., 2018b; Fierer, Bradford & Jackson, 2007; She et al., 2018; Yao et al., 2017). Although no significant effect of N enrichment on fungal community composition was observed (Fig. 4; Table S5), regression analysis also showed that the relative abundance of Basidiomycota was negatively correlated with soil N availability (NH4+ and NO3−, Table S6), indicating that N enrichment likely directly affects Basidiomycota via increasing N availability. Overall, the limited responses of fungal richness, diversity, and community composition under N enrichment in our study may be due to increased N loss from soil through denitrification and leaching (Niu et al., 2016).

Compared to many studies analyzing the response of soil microorganisms to N addition, relatively little research has been done on the effects of P enrichment on soil fungi in grasslands and forests (Li et al., 2015; Liu et al., 2013), and much fewer studies using high-resolution technology, such as 16S rRNA sequencing (Cassman et al., 2016; He et al., 2016). The understanding of fungal community responses to elevated P inputs remains limited (Leff et al., 2015). The effect of P addition on soil fungal community composition depends on ecosystem type, soil properties, and nutrient type and dose (Bao et al., 2013; Beauregard et al., 2010; Li et al., 2015; Liu et al., 2013). Previous studies have demonstrated that experimental P fertilization reduces the species richness of arbuscular mycorrhizal fungi, which could increase soil nutrient capture of their hosts in return for plant C resource (Camenzind et al., 2014; Cheng et al., 2013; Liu et al., 2013). However, elevated P input could alleviate P deficiency of soil microbes and increase the fungal biomass, suggesting P availability is one of the limiting factors for fungal growth in an old tropical forest (Liu et al., 2012a). Another explanation for the not significant change of fungal community composition in response to P enrichment might be the relatively short-term experimental duration of P addition (Cheng et al., 2013), and these detailed parameters on fungal community composition warrant further investigation.

Mechanisms underlying responses of soil bacterial and fungal communities to N and P inputs

Several mechanisms behind the observed shift of bacterial and fungal community composition under N and P additions have been proposed, including increased N and P availability (Liu et al., 2012a, 2013; Nie et al., 2018; Zhou et al., 2017), a decline in soil pH (Chen et al., 2015; Zeng et al., 2016), and increased soil acid cations availability (Chen et al., 2015). Contrary to the N and P availability theory (Liu et al., 2012a; Nie et al., 2018), our results showed that N and P availability (including NH4+, NO3−, and AP) had no significant relationship with bacterial and fungal diversity and community structure (Tables 1 and 2). Meanwhile, both partial Mantel test and SEM results have demonstrated that N and P availability had no significant relationship with bacterial and fungal community composition (Table 2; Fig. 5), suggesting that N and P availability could not explain the shifts in bacterial and fungal community composition. The remaining possible reasons for the decline of bacterial diversity and shift of bacterial community composition include increased concentration of H+ and Al3+ (Table S2). Furthermore, SEM results showed that soil acid cations (H+ and Al3+) induced a significant decrease in bacterial diversity and substantial change of bacterial community composition (Figs. 5 and 6). However, SEM results have demonstrated that soil nutrients are significantly correlated to fungal community composition (Fig. 5). Taken together, in this study, increase of soil acid cations and soil nutrients significantly contributed to the shift in bacterial and fungal community composition, respectively.

Conclusions

After a 5-year N and P addition in a subtropical forest, our results showed that N addition significantly decreased bacterial species richness and diversity, and resulted in a substantial shift of bacterial community composition, whereas P addition did not. Neither N nor P addition changed fungal species richness, diversity, and fungal community composition. A structural equation model showed that the shift in bacterial community composition is attributable to an increase in soil acid cations. The principal component scores of soil nutrients showed a significantly positive relationship with fungal community composition. Our results show how the diversity of microbial communities of subtropical forest soil will depend on future scenarios of anthropogenic N deposition and P enrichment, with a particular sensitivity of bacterial community to N addition.

Supplemental Information

Supplemental Information 1 Supplemental Tables and Figures.

Click here for additional data file.

The authors would like to thank the staff from Jigongshan Natural Reserve for experimental site setup and data collection. Two reviewers and academic editor Xavier Le Roux are appreciated for their constructive comments and valuable input.

Additional Information and Declarations

Competing Interests

Author Contributions

Data Availability

The authors declare that they have no competing interests.

Yong Li conceived and designed the experiments, performed the experiments, analyzed the data, prepared figures and/or tables, authored or reviewed drafts of the paper, approved the final draft.

Dashuan Tian contributed reagents/materials/analysis tools.

Jinsong Wang contributed reagents/materials/analysis tools.

Shuli Niu conceived and designed the experiments, contributed reagents/materials/analysis tools.

Jing Tian contributed reagents/materials/analysis tools, prepared figures and/or tables.

Denglong Ha performed the experiments.

Yuxi Qu performed the experiments.

Guangwei Jing performed the experiments.

Xiaoming Kang authored or reviewed drafts of the paper.

Bing Song analyzed the data, authored or reviewed drafts of the paper.

The following information was supplied regarding data availability:

The raw sequencing data are available at the Sequence Read Archive (SRA): PRJNA531787.

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
