# Peer review of "Differential mechanisms underlying responses of soil bacterial and fungal communities to nitrogen and phosphorus inputs in a subtropical forest"

_PeerJ, doi:10.7717/peerj.7631_

## Round 0.1 · original submission · Minor Revisions

Dear authors,

Both reviewers and myself found that your study has merits and deserves publication in PeerJ, but suggested a number of - most of them minor - adjustments/complements.

Please take all comments into account during the revision process. Would you disagree with a suggestion, please clearly indicate why.
Note that one of the reviewer has attached an annotated pdf; and that I have also attached an annotated word file, both including comments and suggestions that you will have to consider.

Looking forward to receiving the revised version.

best regards,
Xavier

·

Basic reporting

no comment

Experimental design

no comment

Validity of the findings

no comment

Additional comments

This MS detected the effects of N and P addition on soil microbial communities in a subtropical forest. As noted by authors, there was a wide and broad research regarding this topic in grassland ecosystems while was relative lack of research in subtropical forest. Unlike arid or semi-arid grasslands, they found that: 1) N addition decreased bacterial species richness and diversity and resulted in a substantial shift of bacterial community composition, whereas P addition did not; 2) Neither N nor P addition changed fungal species richness, diversity, and fungal community composition; and 3) the shift in bacterial community composition was attributable to an increase in soil acid cations while the shift in fungal community composition was attributable to soil nutrients. The experiment design is reasonable, statistical analysis is ok, and English write is fine. The topic seems suitable for the Journal of PeerJ. Here, I have some minor issues regarding this MS.
1. Line 22-23. The appearance of this sentence is abrupt because you did not mention any pathways in precious sentences at all.
2. Line 257-258. “whereas pH (17.0%), SOC (19.1%), and MBC (41.4%) collectively contributed to 65.2% of the variation”, this sentence for which parameter? Fungal diversity?
3. Line 280. 0.05% should be 5%.
4. One suggestion is that you should cooperate the SEM with Stepwise regression (determinations for alpha-diversity) or partial Mantel test (determinations for beta-diversity) in discussion section. Because all these methods were used to detect the determinations for diversity.

Reviewer 2 ·

Basic reporting

This is a solidly done piece of research and very novel in terms of offering new understanding of the processes involved. I am very impressed with the quality of this manuscript these are thorough analyses, thoughtful discussion points. I think this would make a very good contribution to the journal.

Experimental design

This manuscript explored the responses of soil bacterial and fungal communities and soil parameters to five-year nitrogen and phosphorus addition in a subtropical forest. The authors measured the differences in soil bacterial and fungal communities under N and/or P addition compared to control.

Validity of the findings

Some interesting results were found: (1) N and P addition both increased soil Al3+ and only N addition decreased soil pH; (2) Only N addition decreased bacterial species richness and diversity and neither N nor P addition shifted fungal species richness, diversity and community structure; and (3) SEM result showed that shift in bacterial community composition was attributable to increase in soil acid cations (Al3+) and fungal community composition was closely related with soil nutrients (SOC, TN, DOC), suggesting different ways in which N and P addition influence bacterial and fungal community.

The statistical analysis (i.e., NMDS, ANOSIM, ADONIS, MRPP and SEM) is suitable for the data and provide a sound support for the conclusion. The topic is meaningful for the region and provides some important ecological evidence on responses of bacterial and fungal communities to NP input, which will attract the readers of Peer J.

Additional comments

There are some concerns, which need to be addressed before the paper can be finally accepted for publication.
1. L21. Change “ecosystem function” to “ecosystem functions”
2. L35-36. Change “our results suggest that … in the subtropical forest in different ways.” To “… in different ways in subtropical forest”
3. L58. Delete functional traits
4. L93. Change “does not” to “did not”
5. L303-304. There is a mistake “copiotrophic phylum (Saccharibacteria), oligotrophic phylum (Saccharibacteria)”. Chloroflex should be oligotrophic
6. L392. Change “cause” to “induced”
7. L395-397. This sentence “Take together….shift in bacterial and fungal community composition” could mislead the reader. Please revise it.
8. Line 400. Delete “experiment”.
9. Line 401. Change “decreases” to “decreased”.
10. Captain of Fig. 6, “fungal diversity (c-d)”.

Annotated reviews are not available for download in order to protect the identity of reviewers who chose to remain anonymous.

---

## Round 0.2 · accepted · Accept

Dear authors,

I am pleased to inform you that, following the revision made based on the reviewer’s comments, your manuscript is now acceptable for publication in PeerJ.

Best regards

Xavier LE ROUX